# Probabilistic Margins for Instance Reweighting in Adversarial Training

**Qizhou Wang**[1,*], **Feng Liu**[2,*], **Bo Han**[1,†], **Tongliang Liu**[3], **Chen Gong**[4,5],
**Gang Niu**[6], **Mingyuan Zhou**[7], **Masashi Sugiyama**[6,8]

[1]Department of Computer Science, Hong Kong Baptist University
[2]DeSI Lab, Australian Artificial Intelligence Institute, University of Technology Sydney
[3]TML Lab, School of Computer Science, Faculty of Engineering, The University of Sydney
[4]PCA Lab, Key Lab of Intelligent Perception and Systems for High-Dimensional Information of MoE
[5]Jiangsu Key Lab of Image and Video Understanding for Social Security,
School of Computer Science and Engineering, Nanjing University of Science and Technology
[6]RIKEN Center for Advanced Intelligence Project (AIP)
[7]McCombs School of Business, The University of Texas at Austin
[8]Graduate School of Frontier Sciences, The University of Tokyo
{csqzwang, bhanml}@comp.hkbu.edu.hk, feng.liu@uts.edu.au,
tongliang.liu@sydney.edu.au, chen.gong@njust.edu.cn, gang.niu.ml@gmail.com
mingyuan.zhou@mccombs.utexas.edu, sugi@k.u-tokyo.ac.jp

## Abstract

Reweighting adversarial data during training has been recently shown to improve adversarial robustness, where data *closer* to the current decision boundaries are regarded as *more critical* and given *larger weights*. However, existing methods measuring the closeness are not very reliable: they are *discrete* and can take only a few values, and they are *path-dependent*, i.e., they may change given the same start and end points with different attack paths. In this paper, we propose three types of *probabilistic margin* (PM), which are *continuous* and *path-independent*, for measuring the aforementioned closeness and reweighting adversarial data. Specifically, a PM is defined as the difference between two *estimated class-posterior probabilities*, e.g., such a probability of the true label minus the probability of the most confusing label given some natural data. Though different PMs capture different *geometric properties*, all three PMs share a negative correlation with the vulnerability of data: data with larger/smaller PMs are safer/riskier and should have smaller/larger weights. Experiments demonstrated that PMs are reliable and PM-based reweighting methods outperformed state-of-the-art counterparts.

## 1 Introduction

Deep neural networks are susceptible to adversarial examples that are generated by changing natural inputs with malicious perturbation [17, 20, 37, 39]. Those examples are imperceptible to human eyes but can fool deep models to make wrong predictions with high confidence [3, 35]. As deep learning has been deployed in many real-life scenarios and even safety-critical systems [15, 16, 28], it is crucial to make such deep models reliable and safe [19, 25, 30]. To obtain more reliable deep models, *adversarial training* (AT) [2, 10, 29] was proposed as one of the most effective methodologies against adversary attacks (i.e., maliciously changing natural inputs). Specifically, during training, it simulates adversarial examples (e.g., via *project gradient descent* (PGD) [1, 29, 46, 41]) and train a classifier

---

*Equal contribution.
†Corresponding author.

35th Conference on Neural Information Processing Systems (NeurIPS 2021).

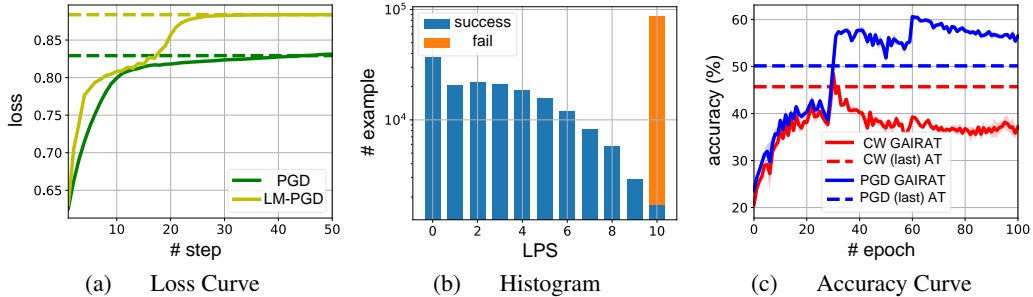

|     |     |     |
|:---:|:---:|:---:|
| (a)   Loss Curve | (b)   Histogram | (c)   Accuracy Curve |

Figure 1: An illustration for the drawbacks of LPS. (a) suggested that LPS is path-dependent and the PGD method (PGD) may get stuck at sub-optimal points. Here, we proved the existence of a better (not optimal) solution given by a LM-PGD method (Appendix A). LPSs are 50 and 14 estimated by PGD and LM-PGD, respectively, which reveals that LPS depends on the adopted attacks. (b) suggested that LPS can take only a few discrete values, and it may have confusing meanings when LPS met its maximum. In this case, one can hardly distinguish non-robust data with LPS being 10 and safe data that are insensitive to be attacked. (c) showed that the accuracy of GAIRAT (on *CIFAR-10*) dropped when facing CW attacks. The main reason is that LPS is not reliable in measuring the distance, and thus it makes wrong judgments when assigning weights for non-robust instances.

with the simulated adversarial examples [29, 46]. Since such a model has seen some adversarial examples during its training process, it can defend against certain adversarial attacks and is more *adversarial-robust* than traditional classifiers trained with natural data [7, 14, 36, 49].

Recently, researchers have found that over-parameterized deep networks still have the insufficient model capacity, due to the overwhelming smoothing effect of AT [47, 33]. As a result, they proposed *instance-reweighted adversarial training*, where adversarial data should have unequal importance given limited model capacity [47]. Concretely, they suggested that data *closer* to the decision boundaries are much more vulnerable to be attacked [45, 47] and should be assigned larger weights during training. To characterize these *geometric properties* of data (i.e., the closeness between the data and decision boundaries), Zhang et al. [47] suggested an estimation in the input space, i.e., the *least PGD steps* (LPS), to identify non-robust (i.e., easily-be-attacked) data. Specifically, LPS is the number of steps to make the adversarial variant of such an instance cross the decision boundaries, starting from a natural instance. Based on LPS, they achieved state-of-the-art performance via a general framework termed *geometry-aware instance-reweighted adversarial training* (GAIRAT).

However, existing methods [18, 22, 38] in measuring the geometric properties of data are *path-dependent*, i.e., they may change even given the same start (a natural example) and end point (the adversarial variant); and they are *discrete* with only a few valid values (Figure 2). The path-dependency makes the computation unstable, where the results may change given different attack paths. The discreteness makes the measurement ambiguous since each value would have several (even contradictory) meanings. We take LPS as an example to demonstrate a path-dependent (Figure 1a) and discrete measurement (Figure 1b). As we can see in Figure 1c, the consequence is non-negligible: although adopting LPS (i.e., GAIRAT) reveals improvement for the PGD-100 attack [29], its performance is well below AT regarding the CW attack [9].

In this paper, we propose the *probabilistic margin* (PM), which is *continuous* and *path-independent*, for reweighting adversarial data during AT. PM is a geometric measurement from a data point to the closest decision boundary, following the *multi-class margin* [23] in traditional geometry-aware machine learning. Note that, instead of choosing the input space as LPS, PM is defined by the *estimated class-posterior probabilities* from the model outputs, e.g., such the

Figure 2: LPS and the proposed PM in comparison. The PGD method takes 6 step to find an instance to violate the decision boundary (i.e., LPS equals 6), while PM is continuous to represent this distance.

probability of the true label minus the probability of the most confusing label given some natural data. Therefore, PM is computed in a low-dimensional embedding space with normalization, alleviating the troubles in comparing data from different classes.

The definition of PM is general, where we consider three specifications, namely, $\text{PM}^{\text{nat}}$, $\text{PM}^{\text{adv}}$, and $\text{PM}^{\text{dif}}$. Concretely, $\text{PM}^{\text{nat}}$ and $\text{PM}^{\text{adv}}$ are the PM scores regarding natural and adversarial data, respectively. They assume that the vulnerability of data is revealed by the closeness regarding either the natural data or the adversarial variants. However, the definition of $\text{PM}^{\text{dif}}$ is slightly different, which is the distance of a natural data point to its adversarial variant. $\text{PM}^{\text{dif}}$ is viewed as a conceptual counterpart of LPS, but is critically different since it is continuous and path-independent. In our paper, we verified the effectiveness of $\text{PM}^{\text{nat}}$ and $\text{PM}^{\text{adv}}$ and showed that they can represent the geometric properties of data well. Note that, though these types of PMs depict different geometric properties, they all share a negative correlation with the vulnerability of data—larger/smaller PMs indicate that the corresponding data are safer/riskier and thus should be assigned with smaller/larger weights.

Eventually, PM is employed for reweighting adversarial data during AT, where we propose the *Margin-Aware Instance reweighting Learning* (MAIL). With a non-increased weight assignment function in Eq. (8), MAIL pays much attention to those non-robust data. In experiments, MAIL was combined with various forms of commonly-used AT methods, including traditional AT [29], MART [40], and TRADES [45]. We demonstrated that PM is more reliable than previous works in geometric measurement, irrelevant to the forms of the adopted objectives. Moreover, in comparison with advanced methods, MAIL revealed its state-of-the-art performance against various attack methods, which benefits from our path-independent and continuous measurement.

## 2 Preliminary

### 2.1 Traditional Adversarial Training

For a $K$-classification problem, we consider a training dataset $S = \{(x_i, y_i)\}_{i=1}^n$ independently drawn from a distribution $\mathcal{D}$ and a deep neural network $h(x; \theta)$ parameterized by $\theta$. This deep classifier $h(x; \theta)$ predicts the label of an input data via $h(x; \theta) = \arg\max_k \mathbf{p}_k(x; \theta)$, with $\mathbf{p}_k(x, \theta)$ being the predicted probability (softmax on logits) for the $k$-th class.

The goal of AT is to train a model with a low adversarial risk regarding the distribution $\mathcal{D}$, i.e., $\mathcal{R}(\theta) = \mathbb{E}_{(x,y)\sim\mathcal{D}} \left[ \max_{\delta\in\Delta} \ell(x + \delta, y; \theta) \right]$, where $\Delta$ is the threat model, defined by an $L_p$-norm bounded perturbation with the radius $\epsilon$: $\Delta = \{\delta \in \mathbb{R}^d \mid ||\delta||_p \leq \epsilon\}$. Therein, AT computes the new perturbation to update the model parameters, where the PGD method [29] is commonly adopted: for a (natural) example $x_i$, it starts with random noise $\xi$ and repeatedly computes

$$\delta_i^{(t)} \leftarrow \texttt{Proj} \left[ \delta_i^{(t-1)} + \alpha\texttt{sign} \left( \nabla_\theta \ell(x_i + \delta_i^{(t-1)}, y_i; \theta) \right) \right], \tag{1}$$

with $\texttt{Proj}$ the clipping operation such that $\delta^{(t)}$ is always in $\Delta$ and $\texttt{sign}$ the signum function. Due to the non-convexity, we typically approximate the optimal solution by $\delta_i^{(T)}$ with $T$ being the maximally allowed iterations. Accordingly, $\delta_i^{(T)}$ is viewed as the perturbation for the *most* adversarial example [47], and the learning objective function is formulated by

$$\arg\min_\theta \sum_i \ell(x_i + \delta_i^{(T)}, y_i; \theta). \tag{2}$$

Intuitively, AT corresponds to the *worst-case robust optimization*, continuously augmenting the training dataset with adversarial variants that highly confuse the current model. Therefore, it is a practical learning framework to alleviate the impact of adversarial attacks. Unfortunately, it leads to insufficient network capacity, resulting in unsatisfactory model performance regarding adversarial robustness. The reason is that, AT has an overwhelming smoothing effect in fitting highly adversarial examples [45], and thus consumes large model capacity to learn from some individual data points.

### 2.2 Geometry-Aware Adversarial Training

Zhang et al. [47] claimed that training examples should have unequal significance in AT, and proposed the *geometry-aware instance-reweighted adversarial training* (GAIRAT). It is a general framework

to reweight adversarial data during training, where Eq. (2) is modified as

$$\arg\min_\theta \sum_i \omega_i \ell(x_i + \delta_i^{(T)}, y_i; \theta) \quad \text{s.t. } \omega_i \geq 0 \text{ and } \sum_i \omega_i = 1. \tag{3}$$

Note that, the constraints are required since the risk after weighting is consistent with the original one without weighting. Further, the generation of perturbation still follows Eq. (1). They revealed that data near decision boundary are much vulnerable to be attacked and require large weights.

**LPS as a geometric measurement.** In assigning weights, GAIRAT needs a proper measurement for the distance to the decision boundaries. They suggested the estimation in high-dimensional input space via LPS (Fig. 2), which is the least PGD iterations for a perturbation that leads to a wrong prediction. Intuitively, a small LPS indicates that the data point can quickly cross the decision boundary and thus close to it.

**The drawbacks of LPS.** Although promising results have been verified in experiments, LPS is path-dependent and limited by a few discrete values, where the consequence is non-negligible. In Figure 1(a), we showed that the PGD method will get stuck. Therefore, using LPS as a geometric measurement, we might identify non-robust examples as robust ones. Here, we modified the vanilla PGD method with the *line-searched* learning rate [43] and *Nesterov momentum* [31] (see Appendix A), termed *Line-search & Momentum-PGD* (LM-PGD), and we compared the loss curve of PGD with that of LM-PGD on one example. The maximal iterations of PGD was 50. As we can see, PGD almost converged at the 14-th step, and the loss value did not ascend anymore. As a result, LPS of this example was 50, and this example was taken as a robust one in the view of LPS. However, LM-PGD still ascended after the 14-th step and successfully attacked the instance at the 15-th step. This result means that such an instance is *not* a robust one, but LPS made a wrong judgment in its robustness. The main reason is that, LPS is heavily dependent on attack paths, even though both paths are highly similar.

Now, we demonstrate that the limited range of LPS would cause problems as well. In Figure 1b, we show the histogram of LPSs for data on CIFAR-10 [24]. Higher LPS values mean that these examples were more robust and required smaller weights during AT. It could be seen that LPS has a confusing meaning when it equaled the maximal value, which is 10 following [29]. For data whose LPSs were 10, they would be the most robust/safe ones. However, it could be seen that they still contained the critical data points (the blue part). Although the proportion of the critical data seems low, ignoring them during AT (i.e., assigning small weights for them) will cause problems. For example, the trained classifier's accuracy will drop significantly when facing the CW attack [9] (Figure 1c).

## 3 Probabilistic Margins for Instance Reweighting

The drawbacks of LPS motivate us to improve the measurement in discerning robust data and risky data, and we introduce our proposal in this section.

### 3.1 Geometry Information in view of Probabilistic Margin

Instead of using the input space as LPS, we suggest the measurement on estimated class-posterior probabilities, which are normalized embedding features (softmax on logits) in the range $[0, 1]$ for each dimension. Note that, without normalization, average distances from different classes might be of diverse scales (e.g., the average distance is 10 for the $i$-th class and 100 for the $j$-th class), increasing the challenge in comparing data from different classes (see Appendix B).

Inspired by the *multi-class margin* in margin theory [23], we propose the *probabilistic margin* (PM) regarding model outputs, namely,

$$\texttt{PM}(x, y; \theta) = \mathbf{p}_y(x; \theta) - \max_{j, j \neq y} \mathbf{p}_j(x; \theta), \tag{4}$$

where the first term in the r.h.s. is the *closeness* of $x$ to the "center" of the true label $y$ and the second term is the closeness to the nearest class except $y$ (i.e., the most confusing label). The difference between the two terms is clearly a valid measurement, where the magnitude reflects the distance from the nearest boundary, and the signum indicates which side the data point belongs to. Figure 3

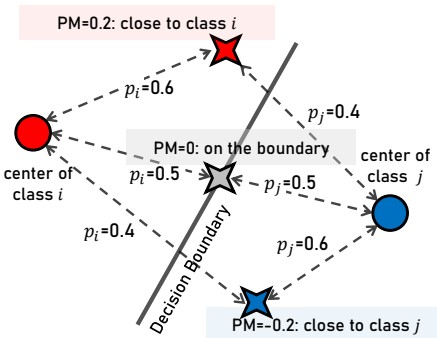

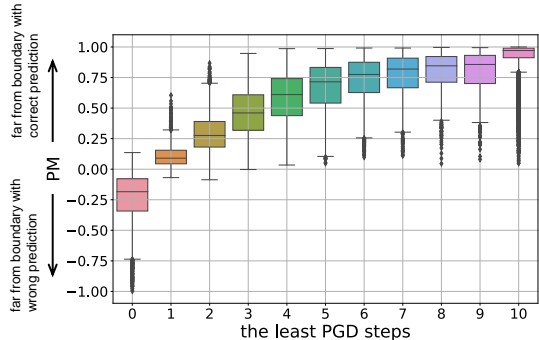

Figure 3: An illustration of PM, where $p_i$ is the probability that a data point belongs to the $i$-th class. As a special example, here we assume PM $= p_i - p_j$ with $i$ the target $y$ and $j$ the nearest class except $y$.

Figure 4: Comparison of LPS and PM (PM$^{\text{nat}}$). An interquartile range box represents the middle 50% of the considered data with the inside line being the median; a whisker is the range for the bottom/top 5% data, and points above/below whiskers are outliers.

summarizes the key concepts, with $i$ for the true label $y$ and $j$ for the most confusing class. For example, when $\mathbf{p}_y(x;\theta) = \mathbf{p}_j(x;\theta) = 0.5$, PM is 0 and the data point $x$ is on the decision boundary between two classes; when $\mathbf{p}_y(x;\theta) = 0.6$ and $\mathbf{p}_j(x;\theta) = 0.4$, PM is positive and the data point $x$ is much closer to the true label; if $\mathbf{p}_y(x;\theta) = 0.4$ and $\mathbf{p}_j(x;\theta) = 0.6$, PM is negative and the data point $x$ is much closer to the most confusing class.

The above discussion indicates that PM can point out which geometric area a data point belongs to, where we discuss the following three scenarios: the *safe area* with large positive PMs, the *class-boundary-around data* with positive PMs close to 0, and the *wrong-prediction area* with negative PMs. The safe area contains guarded data that are insensitive to perturbation, which are safe and require low attention in AT (i.e., small weights); for the class-boundary-around data, they are much vulnerable to be attacked [45, 47] and thus need larger weights than the safe data; for data in the wrong-prediction area, they are the most critical since the attack method can successfully fool the current model, and thus they should be assigned the largest weights. For a data point, this indicates a negative correlation of its PM with the vulnerability, where a larger PM indicates a smaller weight is required for this data point. In realization, the measurement of PM can be employed for adversarial data or natural ones, which are of the forms:

$$\text{PM}_i^{\text{adv}} = \mathbf{p}_{y_i}(x_i + \delta_i^{(T)}; \theta) - \max_{j, j \neq y_i} \mathbf{p}_j(x_i + \delta_i^{(T)}; \theta), \tag{5}$$

$$\text{PM}_i^{\text{nat}} = \mathbf{p}_{y_i}(x_i; \theta) - \max_{j, j \neq y_i} \mathbf{p}_j(x_i; \theta), \tag{6}$$

for a data point $x_i$ respectively. They assume that the vulnerability of data is revealed by the closeness regarding either the natural data or their adversarial variants. Besides these two basic cases, one can also consider the difference between the natural and adversarial predictions, namely,

$$\text{PM}_i^{\text{dif}} = \mathbf{p}_{y_i}(x_i; \theta) - \mathbf{p}_{y_i}(x_i + \delta_i^{(t_i)}; \theta), \tag{7}$$

where $t_i \leq T$ denotes LPS of $x_i$. PM$^{\text{dif}}$ is a conceptual counterpart of LPS, while Eq. (7) is actually path-independent and continuous, which is more reliable than LPS. Note that, the valid range of PM$^{\text{dif}}$ (i.e., $[0, 1]$) is different from that of PM$^{\text{adv}}$ and PM$^{\text{nat}}$ (i.e., $[-1, 1]$), which may bring unnecessary troubles since the geometric meanings of PM$^{\text{nat}}$ and PM$^{\text{dif}}$ are highly similar. Therefore, in our experiments, we mainly verify the effectiveness of PM$^{\text{adv}}$ and PM$^{\text{nat}}$.

Without direct involvement of the PGD method, PM is continuous and path-independent, making it more reliable than LPS. In Figure 4, we depicted the box plot regarding PM for training data with various LPSs. The instability of LPS is evident: from the box centers, there is a little differentiation for data with large LPSs (e.g., LPS = 7 or 9) regarding PM; from the whiskers and outliers, the spreads of PMs are relatively scattered and the numbers of outliers are significant, confirming that the geometric messages characterized by LPS may not be very stable.

---

**Algorithm 1** MAIL: The Overall Algorithm.

---

**Input:** a network model with the parameters $\theta$; and a training dataset $S$ of size $n$.
**Output:** a robust model with parameters $\theta^*$.
 1: **for** $e = 1$ **to** `num_epoch` **do**
 2:   **for** $b = 1$ **to** `num_batch` **do**
 3:     sample a mini-batch $\{(x_i, y_i)\}_{i=1}^m$ from $S$;                  ▷ mini-batch of size $m$.
 4:     **for** $i = 1$ **to** `batch_size` **do**
 5:       $\delta_i^{(0)} = \xi$, with $\xi \sim \mathcal{U}(0, 1)$;
 6:       **for** $t = 1$ **to** $T$ **do**
 7:         $\delta_i^{(t)} \leftarrow \texttt{Proj}\left[\delta_i^{(t-1)} + \alpha\,\texttt{sign}\left(\nabla_\theta \ell(x_i + \delta_i^{(t-1)}, y_i; \theta)\right)\right]$;
 8:       **end for**
 9:       $w_i^{\text{unn}} = \texttt{sigmoid}(-\gamma(\text{PM}_i - \beta))$;
10:     **end for**
11:     $\omega_i = M \times w_i^{\text{unn}} / \sum_j w_j^{\text{unn}}, \forall i \in [m]$;            ▷ $\omega_i = 1$ during burn-in period.
12:     $\theta \leftarrow \theta - \eta\nabla_\theta \sum_{i=1}^m \omega_i \ell(x_i + \delta_i, y_i; \theta) + \mathcal{R}(x_i, y_i; \theta)$;
13:   **end for**
14: **end for**

---

## 3.2 Margin-Aware Instance Reweighting Learning (MAIL)

To benchmark our proposal against state-of-the-art counterparts, we propose the *margin-aware instance reweighting learning* (MAIL). The overall algorithm is summarized in Algorithm 1. Generally, the objective is $\sum_i \omega_i \ell(x_i + \delta_i^{(T)}, y_i; \theta) + \mathcal{R}(x_i, y_i; \theta)$, where $\mathcal{R}$ is an optional regularization term. This objective implies the optimization for the model, with one step (Step 5–8) generating the adversarial variants, one step (Step 9–11) calculating the importance weights, and one step (Step 12) minimizing the reweighted loss w.r.t. the model parameters.

**Weight Assignment:** We adopt the sigmoid function for weight assignment, which can be viewed as a *softened* sample selection operation of the form:

$$\omega_i^{\text{unn}} = \texttt{sigmoid}(-\gamma(\text{PM}_i - \beta)), \tag{8}$$

where $\beta$ indicates how many data should have relatively large weights and $\gamma \geq 0$ controls the smoothness around $\beta$. Note that, $\text{PM}_i$ denotes the PM score for the $i$-th data point, which could be any one of $\text{PM}_i^{\text{adv}}$, $\text{PM}_i^{\text{nat}}$, and $\text{PM}_i^{\text{dif}}$. Eq. (8) is a monotonic function that assigns large values for data with small PMs, paying attention to critical data as discussed in Section 2.2. Moreover, it should be further normalized by $\omega_i = N \times w_i^{\text{unn}} / \sum_j w_j^{\text{unn}}$ to meet the constraint in Eq. (3).

**Burn-in Period:** During the initial training phase, the geometric information is less informative since the deep model is not adequately learned. Directly using the computed weights may mislead the training procedure and accumulate the bias in erroneous weight assignment. Therefore, we introduce a burn-in period at the beginning, where $\omega$ is fixed to 1 regardless of the corresponding PM value. A similar strategy has also been considered in Zhang et al. [47].

**Two Realizations:** The proposed MAIL is general for reweighting adversarial data, which can be combined with existing works. Here we give two representative examples: the first one is based on the vanilla AT [40], with the learning objective of the form (termed MAIL-AT):

$$-\sum_i \omega_i \log \mathbf{p}_{y_i}(x_i + \delta_i^{(T)}; \theta). \tag{9}$$

The second one is based on TRADES [45], which adopts the Kullback-Leibler (KL) divergence regarding natural and adversarial prediction, and also requires the learning guarantee (taken as a regularization term here) on the natural prediction. Overall, the learning objective is (termed MAIL-TRADES)

$$\beta \sum_i \omega_i \texttt{KL}(\mathbf{p}(x_i + \delta_i^{(T)}; \theta) || \mathbf{p}(x_i; \theta)) - \sum_i \log \mathbf{p}_{y_i}(x_i; \theta), \tag{10}$$

where $\beta > 0$ is the trade-off parameter and $\texttt{KL}(p||q) = \sum_k p_k \log p_k/q_k$ denotes the KL divergence.

Table 1: Comparison of LPS and PM on CIFAR-10 dataset.

| | AT [29] | | | MART [40] | | | TRADES [45] | | |
|---|---|---|---|---|---|---|---|---|---|
| | LPS | $PM_{nat}$ | $PM_{adv}$ | LPS | $PM_{nat}$ | $PM_{adv}$ | LPS | $PM_{nat}$ | $PM_{adv}$ |
| NAT | 82.26 ± 0.60 | 82.50 ± 0.20 | **83.15** ± **0.46** | **84.45** ± **0.17** | 83.75 ± 0.09 | 84.12 ± 0.46 | 78.52 ± 0.20 | **82.71** ± **0.52** | 80.49 ± 0.18 |
| PGD | 54.14 ± 0.15 | 55.00 ± 0.32 | **55.25** ± **0.23** | 53.16 ± 0.26 | 53.63 ± 0.21 | **53.65** ± **0.23** | 51.67 ± 0.54 | 52.37 ± 0.65 | **53.67** ± **0.14** |
| AA | 36.32 ± 0.57 | **44.25** ± **0.45** | 44.10 ± 0.21 | 46.00 ± 0.26 | 46.70 ± 0.10 | **47.20** ± **0.21** | 44.40 ± 0.20 | 49.52 ± 0.11 | **50.60** ± **0.22** |

Our proposal is flexible and general enough in combining with many other advanced methods. For example, we can modify MART [40], which could discern correct/wrong prediction, to further utilize their geometric properties (termed MAIL-MART). Its formulation is similar to that of MAIL-TRADES with a slightly different learning objective, which is provided in Appendix C.

## 4 Experiments

We conducted extensive experiments on various datasets, including SVHN [32], CIFAR-10 [24], and CIFAR-100 [24]. The adopted backbone models are ResNet (ResNet-18) [21] and wide ResNet (WRN-32-10) [44]. In Section 4.2, we verified the effectiveness of PM as a geometric measurement. In Section 4.3, we benchmarked our MAIL against advanced methods. The source code of our paper can be found in github.com/QizhouWang/MAIL.

### 4.1 Experimental Setup

**Training Parameters.** For the considered methods, networks were trained using mini-batch gradient descent with momentum 0.9, weight decay $3.5 \times 10^{-3}$ (for ResNet-18) / $7 \times 10^{-4}$ (for WRN-32-10), batch size 128, and initial learning rate 0.01 (for ResNet-18) / 0.1 (for WRN-32-10) which is divided by 10 at the 75-th and 90-th epoch. To some extent, this setup can alleviate the impact of adversarial over-fitting [33, 40]. Moreover, following Madry et al. [29], the perturbation bound $\epsilon$ is $8/255$ and the (maximal) number of PGD steps $k$ is 10 with step size $\alpha = 2/255$.

**Hyperparameters.** The slope and bias parameters were set to 10 and $-0.5$ in MAIL-AT and to 2 and 0 in both MAIL-TRADES and MAIL-MART. The trade-off parameter $\beta$ was set to 5 in MAIL-TRADES, and to 6 in MAIL-MART (Algorithm 4 in Appendix C). For the training procedure, the weights started to update when the learning rate drop at the first time following Zhang et al. [47], i.e., the initial 74 epochs is burn-in period, and then we employed the reweighted objective functions.

**Robustness Evaluation.** We evaluated our methods and baselines using the standard accuracy on natural test data (NAT) and the adversarial robustness based on several attack methods, including the PGD method with 100 iterations [29], CW attack [8], APGD CE attack (APGD) [13], and auto attack (AA) [13]. All these methods have full access to the model parameters (i.e., *white-box* attacks) and are constrained by the same perturbation limit as above. Note that, here we do not focus on the *black-box* attack methods [4, 12, 26, 27, 48], which are relatively easy to be defended [11].

### 4.2 Effectiveness of Probabilistic Margin

In this section, we verified the effectiveness of PM as a measurement in comparison with LPS. Here, we adopted ResNet-18 as the backbone model and conducted experiments on CIFAR-10 dataset.

Three basic methods were considered, including AT [29], MART [40], and TRADES [45], which were further assigned weights given by either PM or LPS [47]. For LPS, we adopted the assignment function in GAIRAT with the suggested setup [47]. We remind that AT-LPS, MART-LPS, and TRADES-LPS represent GAIRAT, GAIR-MART, and GAIR-TRADES in [47]; and AT-PM, MART-PM, and TRADES-PM represent MAIL-AT, MAIL-MART, and MAIL-TRADES, respectively. Note that, we validated two types of PM, namely, $PM^{adv}$ and $PM^{nat}$, which are very different from LPS.

Table 2: Average accuracy (%) and standard deviation on CIFAR-10 dataset with ResNet-18.

| | NAT | PGD | APGD | CW | AA |
|---|---|---|---|---|---|
| AT [29] | 84.86 ± 0.17 | 48.91 ± 0.14 | 47.70 ± 0.06 | 51.61 ± 0.15 | 44.90 ± 0.53 |
| TRADES [45] | 84.00 ± 0.23 | 52.66 ± 0.16 | 52.37 ± 0.24 | 52.30 ± 0.06 | 48.10 ± 0.26 |
| MART [40] | 82.28 ± 0.14 | 53.50 ± 0.46 | 52.73 ± 0.57 | 51.59 ± 0.16 | 48.40 ± 0.14 |
| FAT [46] | **87.97** ± **0.15** | 46.78 ± 0.12 | 46.68 ± 0.16 | 49.92 ± 0.26 | 43.90 ± 0.82 |
| AWP [42] | 85.17 ± 0.40 | 52.63 ± 0.17 | 50.40 ± 0.26 | 51.39 ± 0.18 | 47.00 ± 0.25 |
| GAIRAT [47] | 83.22 ± 0.06 | 54.81 ± 0.15 | 50.95 ± 0.49 | 39.86 ± 0.08 | 33.35 ± 0.57 |
| MAIL-AT | 84.52 ± 0.46 | **55.25** ± **0.23** | **53.20** ± **0.38** | 48.88 ± 0.11 | 44.22 ± 0.21 |
| MAIL-TRADES | 81.84 ± 0.18 | 53.68 ± 0.14 | 52.92 ± 0.62 | **52.89** ± **0.31** | **50.60** ± **0.22** |

The experimental results with 5 individual trials are summarized in Table 1, where we adopted three evaluations, including natural performance (NAT), the PGD method with 100 steps (PGD), and auto attack (AA). AA can be viewed as an ensemble of several advanced attacks and thus reliably reflect the robustness. As we can see, the superiority of PM is apparent, regardless of the adopted learning objectives: the results of PM are 0.70-7.93% better than LPS regarding AA and 0.14-2.00% better regarding PGD. Although LPS could achieve comparable results regarding PGD attacks, its adversarial robustness is quite low when facing AA attacks. It indicates that the robustness improvement of GAIRAT might be partially. This is probably caused by *obfuscated gradients* [2], since stronger attack methods (e.g., AA) lead to poorer performance on adversarial robustness.

Comparing the results in using $PM_{nat}$ and $PM_{adv}$, both PMs can lead to promising robustness, while $PM_{adv}$ is slightly better. The reason is that $PM_{adv}$ can precisely describe the distance between adversarial variants and decision boundaries. As a result, it can help accurately assign high wights for those important instances during AT. Therefore, we adopt $PM_{adv}$ in the following experiments.

### 4.3 Performance Evaluation

We also benchmarked our proposal against advanced methods. Here, we reported the results on the CIFAR-10 dataset due to the space limitation. Please refer to Appendix B for more results.

**Compared Baselines.** We compared the proposed method with the following baselines: (1) (traditional) AT [29]: the cross-entropy loss for adversarial perturbation generated by the PGD method; (2) TRADES [45]: a learning objective with an explicit trade-off between the adversarial and natural performance; (3) MART [40]: a training strategy which treats wrongly/correctly predicted data separately; (4) FAT [46]: adversarial training with early-stopping in adversarial intensity; (5) AWP [42]: a double perturbation mechanism that can flatten the loss landscape by weight perturbation; (6) GAIRAT [47]: geometric-aware instance-reweighted adversarial training.

All the methods were run for 5 individual trials with different random seeds, where we reported their average accuracy and standard deviation. The results are summarized in Table 2 and Table 3 with the backbone models being ResNet-18 and WRN-32-10, respectively. Overall, our MAIL achieved the best or the second-best robustness against all four types of attacks, revealing the superiority of MAIL (i.e., MAIL-AT and MAIL-TRADES) in adversarial robustness. Specifically, AT and TRADES both treat training data equally, and thus their results were unsatisfactory compared with the best one (0.55% − 6.34% decline with ResNet-18 and 0.02% − 9.43% decline with WRN-32-10), implying the possibility for its further improvement. Though MART and FAT consider the impact

Table 3: Average accuracy (%) and standard deviation on CIFAR-10 dataset with WRN-32-10.

| | NAT | PGD | APGD | CW | AA |
|---|---|---|---|---|---|
| AT | 87.80 ± 0.13 | 49.43 ± 0.29 | 49.12 ± 0.26 | 53.38 ± 0.05 | 48.46 ± 0.46 |
| TRADES | 86.36 ± 0.52 | 54.88 ± 0.39 | 55.02 ± 0.27 | 56.18 ± 0.16 | 53.40 ± 0.37 |
| ' MART | 84.76 ± 0.34 | 55.61 ± 0.51 | 55.40 ± 0.37 | 54.72 ± 0.20 | 51.40 ± 0.05 |
| FAT | **89.70 ± 0.17** | 48.79 ± 0.18 | 48.72 ± 0.36 | 52.39 ± 0.89 | 47.48 ± 0.30 |
| AWP | 57.55 ± 0.23 | 54.17 ± 0.10 | 54.20 ± 0.16 | 55.18 ± 0.30 | 53.08 ± 0.17 |
| GAIRAT | 86.30 ± 0.61 | 58.74 ± 0.46 | 55.64 ± 0.36 | 45.57 ± 0.18 | 40.30 ± 0.16 |
| MAIL-AT | 84.83 ± 0.39 | **58.86 ± 0.25** | 55.82 ± 0.31 | 51.26 ± 0.20 | 47.10 ± 0.22 |
| MAIL-TRADES | 84.00 ± 0.15 | 57.40 ± 0.96 | **56.96 ± 0.19** | **56.20 ± 0.30** | **53.90 ± 0.22** |

of individuals on the final performance, they fail in paying attention to geometric properties of data during training. Concretely, MART mainly focuses on the correctness regarding natural prediction, and FAT prevents the model learning from *highly* non-robust data in keeping its natural performance. Therefore, FAT achieved the best natural performance ($3.11\% - 6.13\%$ improvement with ResNet-18 and $1.90\% - 5.70\%$ improvement with WRN-32-10), while its robustness against adversaries seems inadequate ($2.97\% - 8.45\%$ decline with ResNet-18 and $3.81\%$-10.07% decline with WRN-32-10). Moreover, in adopting LPS as the geometric measurement, GAIRAT performed well regarding PGD and AGPD attacks, while the adversarial robustness on CW and AA attacks were pretty low. In comparison, we retained the supremacy on PGD-based attacks (i.e., PGD and APGD) as in GAIRAT and revealed promising results regarding CW and AA attacks. For example, in Table 2, we achieved 0.44%-8.47% improvements on PGD attack, 0.47%-6.52% on APGD attack, 0.59%-13.03% on CW attack, and 1.63%-16.68% on AA attack.

MAIL-AT performed well on PGD-based methods, while MAIL-TRADES was good at CW and AA attacks. It suggests that the robustness depends on adopted learning objectives, coinciding with the previous conclusion [40]. In general, we suggest using MAIL-TRADES as a default choice, as it reveals promising results regarding AA while keeping a relatively high performance regarding PGD-based attacks. Besides, comparing the results in Table 2 and Table 3, the overall accuracy had a promising improvement in employing models with a much large capacity (i.e., WRN-32-10), verifying the fact that deep models have insufficient network capacity in fitting adversaries [47].

## 5 Conclusion

In this paper, we focus on boosting adversarial robustness by reweighting adversarial data during training, where data closer to the current decision boundaries are more critical and thus require larger weights. To measure the closeness, we suggest the use of *probabilistic margin* (PM), which relates to the multi-class margin in the probability space of model outputs (i.e., the estimated class-posterior probabilities). Without any involvement of the PGD iterations, PM is continuous and path-independent and thus overcomes the drawbacks of previous works (e.g., LPS) efficaciously. Moreover, we consider several types of PMs with different geometric properties, and propose a general framework termed MAIL. Experiments demonstrated that PMs are more reliable measurements than previous works, and our MAIL revealed its superiority against state-of-the-art methods, independent of adopted (basic) learning objectives. In the future, we will delve deep into the mechanism in instance-reweighted adversarial learning, theoretically study the contribution of individuals for the final performance, and improve the methodology in using geometric characteristics of data.

## Acknowledgments and Disclosure of Funding

QZW and BH were supported by the RGC Early Career Scheme No. 22200720, NSFC Young Scientists Fund No. 62006202 and HKBU CSD Departmental Incentive Grant. TLL was partially supported by Australian Research Council Projects DP-180103424, DE-190101473, and IC-190100031. GN and MS were supported by JST AIP Acceleration Research Grant Number JPMJCR20U3, Japan, and MS was also supported by Institute for AI and Beyond, UTokyo.

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
