# A Line-Search & Momentum-PGD (LM-PGD)

Here, we describe the detailed realization of the Line-Search & Momentum-PGD (LM-PGD) method. Compared with the commonly used PGD method of the form following

$$\delta_i^{(t)} \leftarrow \texttt{Proj}\left[\delta_i^{(t-1)} + \alpha\texttt{sign}\left(\nabla_\theta \ell(x_i + \delta_i^{(t-1)}, y_i; \theta)\right)\right], \tag{11}$$

the Line-Search & Momentum-PGD makes two improvements: (1) we use the Nesterov gradient [31] in an attempt to overcome the sub-optimal points, i.e.,

$$v_i^{(t)} \leftarrow \gamma v_i^{(t-1)} + \alpha^{(t-1)}\texttt{sign}\left(\nabla_\theta \ell(x_i + \delta_i^{(t-1)}, y_i; \theta)\right), \tag{12}$$

$$\delta_i^{(t)} \leftarrow \texttt{Proj}\left[\delta_i^{(t-1)} + v_i^{(t)}\right], \tag{13}$$

where $\gamma$ is the momentum parameter and $\alpha^{(t)}$ is the step size at the $t$-th iteration; (2) and we adopt the line-searched step size [43], which aims at finding the optimal $\alpha^{(t-1)}$ in a predefined searching space $V = [\alpha^{(\min)}, \alpha^{(\max)}]$, namely

$$\alpha^{(t-1)} \leftarrow \arg\max_{\alpha \in V} \ell\left(x_i + \texttt{Proj}\left[\delta_i^{(t-1)} + \gamma v_i^{(t-1)} + \alpha\texttt{sign}\left(\nabla_\theta \ell(x_i + \delta_i^{(t-1)}, y_i; \theta)\right)\right], y_i; \theta\right). \tag{14}$$

In Figure 1(a), the maximal PGD iteration is 50, with $\gamma$ being 0.8, $\alpha^{(\max)}$ being $6/255$, and $\alpha^{(\min)}$ being $4/255$ in the first 15 iterations and 0 otherwise.

Note that, the LM-PGD cannot remove the drawbacks of the LPS with the following reasons: (1) the path-dependency property exists as the LM-PGD still uses the first-order gradients for optimization; (2) the momentum term and the line-searched learning rate would make the corresponding LPS difficult to be computed since each step may have different length. As the LM-PGD cannot be directly used in assigning weights, we proposed the PM in this paper.

# B Further Experiments

To further demonstrate the robustness of our proposal against adversarial attacks, we also conducted experiments on SVHN [32] and CIFAR-100 [32] datasets. Here we adopted ResNet-18 as the backbone model with the same experimental setup as in Section 4.2, where we reported the natural accuracy (NAT), PGD-100 attack (PGD), and auto-attack (AA). All the experiments are conducted for 5 individual trails, and the average performance and standard deviations are summarized in Table 4. Note that, all the methods were realized by Pytorch 1.81 [3] with CUDA 11.1 [4], where we used several GeForce RTX 3090 GPUs and AMD Ryzen Threadripper 3960$X$ Processors.

Some advanced methods (e.g., TRADES and MART) can hardly beat the traditional AT on these two datasets, indicating the performance of the learning methods may also depend on the considered datasets. Moreover, FAT retains the highest natural performance (i.e., NAT) but its results on PGD and AA are relatively low, which are similar to Table 2 and Table 3. For GAIRAT on CIFAR-100, we observe that its performance on the PGD attack was not as high as some other datasets (e.g., SVHN and CIFAR-10), it may reveal that the deficiencies of LPS are non-negligible for some difficult tasks. There may exist some better reweighting strategies in boosting adversarial robustness. As we can see, our MAIL-AT and MAIL-TRADES, which benefit from the proposed PM, achieved the best or the comparable (e.g., AA on CIFAR-100) performance regarding various adversarial attacks, and the natural accuracy retained to be acceptable.

Further, we tested the effectiveness in adopting the probabilistic space, where we compare our PM with the results given by multi-class margin (denoted by MM) of the form:

$$f_{y_i}(x_i; \theta) - \max_{j, j \neq y_i} f_j(x_i; \theta), \tag{15}$$

where $f$ denotes the model outputs before the softmax link function and $f_j$ denotes its $j$-th dimensional value. We conducted experiments on CIFAR-10 dataset with the backbone being ResNet-18

---

[3]https://pytorch.org/
[4]https://developer.nvidia.com/

Table 4: Average test accuracy (%) and standard deviation (5 trials) on SVHN and CIFAR-100.

| | SVHN | | | CIFAR-100 | | |
|---|---|---|---|---|---|---|
| | NAT | PGD | AA | NAT | PGD | AA |
| AT | 93.55 ± 0.65 | 54.76 ± 0.30 | 46.58 ± 0.46 | 60.13 ± 0.39 | 28.69 ± 0.24 | 24.76 ± 0.25 |
| TRADES | 93.65 ± 0.29 | 55.12 ± 0.68 | 45.74 ± 0.31 | 60.73 ± 0.33 | 29.83 ± 0.25 | 24.90 ± 0.33 |
| MART | 92.57 ± 0.14 | 55.45 ± 0.26 | 43.52 ± 0.43 | 54.08 ± 0.60 | 29.94 ± 0.21 | **25.30 ± 0.50** |
| FAT | **93.68 ± 0.31** | 53.81 ± 0.77 | 41.68 ± 0.49 | **66.74 ± 0.28** | 23.25 ± 0.55 | 20.88 ± 0.13 |
| AWP | 93.34 ± 0.22 | 57.62 ± 0.31 | 47.21 ± 0.30 | 55.16 ± 0.27 | 30.14 ± 0.30 | 25.16 ± 0.39 |
| GAIRAT | 90.47 ± 0.77 | 64.67 ± 0.26 | 37.00 ± 0.30 | 58.43 ± 0.28 | 25.74 ± 0.51 | 17.54 ± 0.33 |
| MAIL-AT | **93.68 ± 0.25** | **65.54 ± 0.16** | 41.12 ± 0.30 | 60.74 ± 0.15 | 27.62 ± 0.27 | 22.44 ± 0.53 |
| MAIL-TRADES | 89.68 ± 0.19 | 58.40 ± 0.21 | **49.10 ± 0.22** | 60.13 ± 0.25 | **30.28 ± 0.19** | 24.80 ± 0.25 |

Table 5: Comparison of MM and PM on CIFAR-10 dataset with average test accuracy (%) and standard deviation (5 trails).

| | MM | PM |
|---|---|---|
| NAT | 80.44 ± 0.21 | **81.84 ± 0.18** |
| PGD | 53.58 ± 0.11 | **53.68 ± 0.14** |
| AA | 49.20 ± 0.09 | **50.60 ± 0.22** |

Table 6: Comparison with different weight assignment functions on CIFAR-10 dataset with average test accuracy (%) and standard deviation (5 trails).

| | hinge | step | sigmoid |
|---|---|---|---|
| NAT | 80.17 ± 0.17 | 81.11 ± 0.15 | **81.84 ± 0.18** |
| PGD | 52.23 ± 0.31 | 53.53 ± 0.14 | **53.68 ± 0.14** |
| AA | 43.40 ± 0.22 | 50.30 ± 0.20 | **50.60 ± 0.22** |

and the basic method being TRADES. The experimental results are listed in the Table 5. As we can see, the normalized embedding features (i.e., our PM) would genuinely lead to better results regarding both natural accuracy (NAT) and adversarial robustness (PGD and AA), verifying the effectiveness in using the probabilistic margin (instead of the commonly-used multi-class margin) as the measurement for geometry-aware adversarial training.

Now, we made investigations of the formulations in assigning weights. In Table 6, we further adopted the following two assignment functions, namely, the hinge-shaped function $\max(0, \gamma(\mathrm{PM}_i - \beta))$ and the step function $\begin{cases} \alpha & \text{if } \mathrm{PM}_i > \beta \\ 1 - \alpha & \text{if } \mathrm{PM}_i \leq \beta \end{cases}$ (with $\alpha = 0.2$). We conducted experiments on CIFAR-10 dataset with backbone being ResNet-18 and basic method being TRADES. The results are listed in Table 6. As we can see, compared with these commonly-used assignment functions, the sigmoid function can truly achieve superior performance, coincide with previous observation in [47]. The reason is that, the sigmoid-shaped functions are strictly monotonic and continuous, where the resulting weights can retain the underlying geometric properties of data properly. In comparison, the robustness regarding AA attack given by hinge-shaped function is far below other assignment functions, since it ignore all those data with $\mathrm{PM}_i - \beta < 0$ by assigning weights with value being 0. Further, the results given by the step function is also lower than our sigmoid-shaped function, since it can not distinguish data with different geometric properties if all these data are greater/smaller than $\beta$.

In the end, it is also interesting in adopting other attacks methods in our learning framework, to demonstrate that our proposal is general to most of the attack methods. Here, we adopted PGD (as used in AT), CW, and KL (as used in TRADES) in generating adversarial perturbations, and we used the learning objectives given by AT and TRADES. We conducted experiments on CIFAR-10 dataset with ResNet-18, and the results are listed in Table 7. In general, adopting the KL method would

Table 7: MAIL with different methods in generating adversarial perturbations on CIFAR-10 dataset.

| | AT | | | TRADES | | |
|---|---|---|---|---|---|---|
| | NAT | PGD | AA | NAT | PGD | AA |
| PGD | $84.52 \pm 0.46$ | $55.25 \pm 0.23$ | $\mathbf{44.22 \pm 0.21}$ | $80.83 \pm 0.31$ | $54.84 \pm 0.15$ | $49.90 \pm 0.10$ |
| CW | $81.84 \pm 0.25$ | $\mathbf{65.55 \pm 0.20}$ | $37.20 \pm 0.18$ | $80.33 \pm 0.19$ | $\mathbf{57.55 \pm 0.32}$ | $50.20 \pm 0.25$ |
| KL | $\mathbf{85.33 \pm 0.30}$ | $51.34 \pm 0.14$ | $32.10 \pm 0.22$ | $\mathbf{81.84 \pm 0.18}$ | $53.68 \pm 0.14$ | $\mathbf{50.60 \pm 0.22}$ |

lead to better results on natural accuracy, while adversarial robustness regarding the PGD attack prefers to learn from the perturbation generated by CW. It coincides with the previous conclusion that adversarial training is not adaptive [34]. Further, for the AA attack, TRADES revealed much stable and superior results regarding various methods in generating perturbations, compared with the results given by AT.

## C  Realizations of MAIL

Here, we provide the detailed algorithms for our three realizations, namely, MAIL-AT (in Algorithm 2), MAIL-TRADES (in Algorithm 3), and MAIL-MART (in Algorithm 4). In Algorithm 4, the *boosted cross entropy* (BCE) loss is of the form:

$$\text{BCE}(x_i + \delta_i^{(T)}, y_i; \theta) = -\log \mathbf{p}_{y_i}(x_i + \delta_i^{(T)}) - \log(1 - \max_{k \neq y_i} \mathbf{p}_k(x_i + \delta_i^{(T)})), \qquad (16)$$

where the first term in the r.h.s. is the common cross entropy loss and the second term is a regularization term that that make the prediction much confident. The *mis-classification aware KL* (MKL) term is of the form:

$$\text{MKL}(x_i, \delta_i^{(T)}; \theta) = \text{KL}(\mathbf{p}(x_i + \delta_i^{(T)}; \theta) || \mathbf{p}(x_i; \theta))(1 - \mathbf{p}_{y_i}(x_i; \theta)), \qquad (17)$$

which reweights the KL-divergence by the estimated probability of a correct prediction.

## D  Further Discussion

We adopt an instance-reweighting learning framework and propose PMs in measuring the robustness regarding adversarial attacks. Our PMs are continuous and path-independent, overcoming the deficiency of previous works [47]. The overall method is general and flexible, combined with existing works to boost their primary performance.

Moreover, there is still room for improvement in our approach and related works. This paper mainly focuses on adversarial robustness regarding white-box attacks generated by the first-order gradient-based methods. When employing our MAIL in real-world applications, it may lead to over-confidence regarding many other attacks, e.g., provable attacks [5], black-box attacks [6], and physical attacks [25]. It poses the potential risk and dangers for the resulting systems (e.g., autonomous driving), as it would produce non-robust behavior unexpectedly when encountering unseen types of attacks.

The problems of fairness should also be considered since individuals are not treated equally during training. For data assigned with larger weights, the resulting model would be more robust when encounters similar data during the test. This unfairness problem seems inevitable for a reweighted learning framework, which will interest our further study. Moreover, our method might be more vulnerable to error-prone data (e.g., with label noise) than traditional AT since MAIL would provide abnormal weights (e.g., larger weights than expectation) for these data. This issue would mislead the training procedure in fitting *noise*, which has an apparent negative impact on the model.

---

**Algorithm 2** MAIL-AT: The Overall Algorithm.

---

**Input:** a network model with the parameters $\theta$; and a training dataset $S$ of size $n$.
**Output:** a robust model with parameters $\theta^*$.

 1: **for** $e = 1$ **to** num_epoch **do**
 2:   **for** $b = 1$ **to** num_batch **do**
 3:     sample a mini-batch $\{(x_i, y_i)\}_{i=1}^m$ from $S$;                  ▷ mini-batch of size $m$.
 4:     **for** $i = 1$ **to** batch_size **do**
 5:       $\delta_i^{(0)} = \xi$, with $\xi \sim \mathcal{U}(0,1)$;
 6:       **for** $t = 1$ **to** $T$ **do**
 7:         $\delta_i^{(t)} \leftarrow \texttt{Proj}\left[\delta_i^{(t-1)} + \alpha\texttt{sign}\left(\nabla_\theta - \log \mathbf{p}_{y_i}(x_i + \delta_i^{(t-1)}; \theta)\right)\right]$;
 8:       **end for**
 9:       $w_i^{\text{unn}} = \texttt{sigmoid}(-\gamma(\text{PM}_i - \beta))$;
10:     **end for**
11:     $\omega_i = M \times w_i^{\text{unn}} / \sum_j w_j^{\text{unn}}, \forall i \in [m]$;                  ▷ $\omega_i = 1$ during burn-in period.
12:     $\theta \leftarrow \theta - \eta\nabla_\theta\left(-\sum_{i=1}^m \omega_i \log \mathbf{p}_{y_i}(x_i + \delta_i^{(T)}; \theta)\right)$;
13:   **end for**
14: **end for**

---

**Algorithm 3** MAIL-TRADES: The Overall Algorithm.

---

**Input:** a network model with the parameters $\theta$; and a training dataset $S$ of size $n$.
**Output:** a robust model with parameters $\theta^*$.

 1: **for** $e = 1$ **to** num_epoch **do**
 2:   **for** $b = 1$ **to** num_batch **do**
 3:     sample a mini-batch $\{(x_i, y_i)\}_{i=1}^m$ from $S$;                  ▷ mini-batch of size $m$.
 4:     **for** $i = 1$ **to** batch_size **do**
 5:       $\delta_i^{(0)} = \xi$, with $\xi \sim \mathcal{U}(0,1)$;
 6:       **for** $t = 1$ **to** $T$ **do**
 7:         $\delta_i^{(t)} \leftarrow \texttt{Proj}\left[\delta_i^{(t-1)} + \alpha\texttt{sign}\left(\nabla_\theta\texttt{KL}(\mathbf{p}(x_i + \delta_i^{(t-1)}; \theta)||\mathbf{p}(x_i; \theta))\right)\right]$;
 8:       **end for**
 9:       $w_i^{\text{unn}} = \texttt{sigmoid}(-\gamma(\text{PM}_i - \beta))$;
10:     **end for**
11:     $\omega_i = M \times w_i^{\text{unn}} / \sum_j w_j^{\text{unn}}, \forall i \in [m]$;                  ▷ $\omega_i = 1$ during burn-in period.
12:     $\theta \leftarrow \theta - \eta\nabla_\theta \sum_i \left(\beta\omega_i\texttt{KL}(\mathbf{p}(x_i + \delta_i^{(T)}; \theta)||\mathbf{p}(x_i; \theta)) - \sum_i \log \mathbf{p}_{y_i}(x_i; \theta)\right)$;
13:   **end for**
14: **end for**

---

**Algorithm 4** MAIL-MART: The Overall Algorithm.

---

**Input:** a network model with the parameters $\theta$; and a training dataset $S$ of size $n$.
**Output:** a robust model with parameters $\theta^*$.

 1: **for** $e = 1$ **to** num_epoch **do**
 2:   **for** $b = 1$ **to** num_batch **do**
 3:     sample a mini-batch $\{(x_i, y_i)\}_{i=1}^m$ from $S$;                  ▷ mini-batch of size $m$.
 4:     **for** $i = 1$ **to** batch_size **do**
 5:       $\delta_i^{(0)} = \xi$, with $\xi \sim \mathcal{U}(0,1)$;
 6:       **for** $t = 1$ **to** $T$ **do**
 7:         $\delta_i^{(t)} \leftarrow \texttt{Proj}\left[\delta_i^{(t-1)} + \alpha\texttt{sign}\left(\nabla_\theta - \log \mathbf{p}_{y_i}(x_i + \delta_i^{(t-1)}; \theta)\right)\right]$;
 8:       **end for**
 9:       $w_i^{\text{unn}} = \texttt{sigmoid}(-\gamma(\text{PM}_i - \beta))$;
10:     **end for**
11:     $\omega_i = M \times w_i^{\text{unn}} / \sum_j w_j^{\text{unn}}, \forall i \in [m]$;                  ▷ $\omega_i = 1$ during burn-in period.
12:     $\theta \leftarrow \theta - \eta\nabla_\theta\left(-\sum_{i=1}^m \omega_i\texttt{BCE}(x_i + \delta_i^{(T)}, y_i; \theta) + \beta\texttt{MKL}(x_i, \delta_i^{(T)}; \theta)\right)$;
13:   **end for**
14: **end for**

---