# OpenReview forum: "Probabilistic Margins for Instance Reweighting in Adversarial Training"
_NeurIPS.cc/2021/Conference — NeurIPS 2021 Poster_

### Official Review · Reviewer_k79K · 2021-07-11

**Rating:** 6
**Confidence:** 5

**Summary:**

This paper propose  three types of probabilistic margin (PM) to improve the model robustness. The proposed method can measure the aforementioned closeness and reweight adversarial data.
And some experiments were conducted to validate the effectiveness of the proposed method.

**Limitations And Societal Impact:**

Yes

**Main Review:**

 This paper have following problems:

1.  the paper lacks comparative experiments with the state-of-the-art methods [1]

2.  The robustness improvement is very small. The proposed MAIL-AT achieves lower AA attack accuracy compared with AT [26] in table 2 (44.10% VS 44.90%) .
and MAIL-TRADES only achieves a little higher AA attack accuracy compared with TRADES in Table 3 ( 53.90% VS 53.40%).

3.  The certain baseline of FAT seem to deviate from values reported in prior works [2].

4. The novelty of work is limited. This paper seem to  use multi-class margin in margin theory [20] to conduct adversarial training.

Therefore, I think this work is not qualified enough for NIPS.

[1] Adversarial Weight Perturbation Helps Robust Generalization, (NeurIPS 2020)
[2] Bag of Tricks for Adversarial Training, Pang et al, (ICLR 2021)


**Time Spent Reviewing:**

30 hours

---

> ### Author Response · Authors · 2021-08-09
> **Response to "the novelty of work is limited."**
>
> >The novelty of work is limited. This paper seems to use multi-class margin in margin theory to conduct adversarial training.
>
> First of all, we want to emphasize that we are the first to introduce the margin to conduct adversarial training (AT), which is motivated by a promising research direction of AT: geometric-aware AT. In this paper, we follow this novel research direction and find that the deficiencies of its geometric measurement make the robustness unsatisfactory. To solve the above problem, we are the first to introduce the probabilistic margin (PM) to measure the closeness between data points and the class boundaries. Based on PM, we can assign weights for adversarial data during AT. Since PM addresses the most important issue of the geometric-aware AT methods, it lights up a new road for the geometric-aware AT. This contribution, in the long run, is significant to the field of adversarial training. Compared with the commonly-used multi-class margin, we suggest the measurement in the probabilistic space of the model outputs, which not only enjoys the properties of the margin theory but is also suitable for data comparison and weight assignment.
>
>
>
> Except for the major contributions demonstrated above, we still have the following two contributions: (1) the investigation of phenomenon where LPS can mislead the weighted adversarial training, leading to the inferior results in GAIRAT; and (2) we propose the probabilistic margin that overcomes the shortages in previous works, and our MAIL reveals the better results than the state-of-the-art counterparts.

---

> ### Author Response · Authors · 2021-08-09
> **Response to other comments**
>
> >Q1. The paper lacks comparative experiments with state-of-the-art methods [r1].
>
> A1. Many thanks for your kind advice. We have conducted experiments on CIFAR-10 with the backbone being ResNet-18, and the results are listed in the following:
>
> |             | NAT   | PGD   | APGD  | CW    | AA    |
> |-------------|-------|-------|-------|-------|-------|
> | AWP         | 78.64 | 51.45 | 49.20 | 45.40 | 45.20 |
> | MAIL-AT     | 83.15 | 55.25 | 53.20 | 47.81 | 44.10 |
> | MAIL-TRADES | 81.84 | 53.68 | 52.92 | 52.89 | 50.03 |
>
> Although the AWP can outperform our MIAL-AT regarding the AA attack, its robustness regarding all other attacks is much lower than our results, and the MAIL-TRADES can always outperform the AWP by a large margin. Overall, we conclude that our proposal can achieve superior performance than the AWP.
>
> >Q2. The robustness improvement is very small. The proposed MAIL-AT achieves lower AA attack accuracy compared with AT in table 2 (44.10\% VS 44.90\%), and MAIL-TRADES only achieves a little higher AA attack accuracy compared with TRADES in Table 3 (53.90\% VS 53.40\%).
>
> A2. Thank you for your concerns! In general, our MAIL can achieve superior results regarding most of the attacks. We have conducted five groups of experiments on three different datasets, more than 20 pairs of robustness comparisons (e.g., AT vs. MAIL-AT and TRADES vs. MAIL-TRADES) are provided therein. Among them, for 18 (out of 24) pairs of results, our MAIL outperforms the original robustness by a large margin; and for all other results, our MAIL can retain the comparable results. It indicates that our MAIL can boost the robustness of the model in general.
>
> >Q3. The certain baseline of FAT seems to deviate from values reported in prior works [r2].
>
> A3. Our experiments followed the released code in [r3,r4], which has a different setup compared with [r2] (e.g., a different learning rate scheduler). Since the FAT is susceptible to the adopted setup and can easily overfit, the reported results in our paper may look different from [r2].
>
>
> [r1] Adversarial Weight Perturbation Helps Robust Generalization (NeurIPS 2020).
>
> [r2] Bag of Tricks for Adversarial Training (ICLR 2021).
>
> [r3] Theoretically Principled Trade-off between Robustness and Accuracy (ICML 2019).
>
> [r4] Improving Adversarial Robustness Requires Revisiting Misclassified Examples (ICLR 2020).

---

### Official Review · Reviewer_SNCE · 2021-07-16

**Rating:** 7
**Confidence:** 3

**Summary:**

This paper focuses on reweighting adversarial data in adversarial training problems. Actually, how to determine the sensible weights for each adversarial example matters for the adversarial robustness of the trained model. In this way, this paper proposes a new measurement called probabilistic margin, and employs it into multiple baseline methods. The achieved results seem promising.

**Limitations And Societal Impact:**

Yes

**Main Review:**

By arguing that the current measurement of "closeness to boundary" is discrete and path-dependent, this paper borrows the tool in margin theory called probabilistic margin (PM) as a more accurate proxy. Then the PM is utilized in multiple baseline methods and achieves better performance.

However, the utilization of PM seems sort of plain to me. Using PM seems a natural option for various proxies of measurement of "closeness to boundary". More theoretical insights are needed.

Authors should make further investigations of the formulation of weight assignment. Why do we choose sigmoid in Eq8?

Instructions of empirical choice of PM_adv, PM_nat, PM_diff are still not clear. What can we draw from the analysis?

-----------------------------------------------------

Thanks for the kind response. After the rebuttal, my concerns are well-addressed. Thus I am happy to raise my score.




**Time Spent Reviewing:**

1

---

> ### Author Response · Authors · 2021-08-09
> **Respone to review questions**
>
> >Q1. However, the utilization of PM seems sort of plain to me. Using PM seems a natural option for various proxies of measurement of "closeness to boundary.” More theoretical insights are needed.
>
> A1. Many thanks for your helpful suggestions. First of all, we want to emphasize that we are the first to introduce the margin to conduct adversarial training (AT), which is motivated by a promising research direction of AT: geometric-aware AT. In this paper, we follow this novel research direction and find that the deficiencies of its geometric measurement make the robustness unsatisfactory. To solve the above problem, we are the first to introduce the probabilistic margin (PM) to measure the closeness between data points and the class boundaries. Based on PM, we can assign weights for adversarial data during AT. Since PM addresses the most important issue of the geometric-aware AT methods, it lights up a new road for the geometric-aware AT. This contribution, in the long run, is also important to the field of adversarial training. Compared with the commonly-used multi-class margin, we suggest the measurement in the probabilistic space of the model outputs, which not only enjoys the properties of the margin theory but is also suitable for data comparison and weight assignment.
>
> We also agree with your advice for theoretical insights. However, given that the adversarial data are non-i.i.d. distributed [r2] and are changing during training, the development of the learning theories in the literature is limited. In the future, we will focus on the advanced works associated with the robustness guarantees for AT, and we will generalize the results to our MAIL.
>
>
> >Q2. Authors should make further investigations of the formulation of weight assignment. Why do we choose sigmoid in Eq8?
>
> A2. Similar to previous conclusions in GAIRAT, we also find a sigmoid-shape function would give us promising experimental results. As a support, we adopt the following two assignment functions: (1) a hinge-shape function $\max(0, -\gamma(PM_i-\beta))$ and (2) a step function $\alpha$ if $PM_i > \beta$ and $(1-\alpha)$ if $PM_i \le \beta$ ($\alpha=0.2$). We conducted experiments on CIFAR-10 with the backbone being ResNet-18, and the results are listed in the following:
>
> |         | NAT   | PGD   | APGD  | CW    | AA    |
> |---------|-------|-------|-------|-------|-------|
> | hinge   | 82.07 | 49.82 | 46.02 | 45.74 | 42.34 |
> | step    | 81.19 | 55.08 | 49.66 | 44.95 | 40.92 |
> | sigmoid | 83.15 | 55.25 | 53.2  | 47.81 | 44.10 |
>
> Therein, the upper-bounded value makes the proposed assignment function better than the hinge-shape function, and the sigmoid function’s smooth property makes its better than the step function. As a result, we suggest the use of the sigmoid function for weight assignment.
>
> >Q3. Instructions of empirical choice of PM_adv, PM_nat, PM_diff are still not clear. What can we draw from the analysis?
>
> A3. Many thanks for your kind advice. When the loss landscape of the model is relatively sharp, the PM_diff (and the LPS) will perform worse than the results of PM_nat and PM_adv. In this case, the PM_diff (and the LPS) would produce an unreliable measurement, and this is the reason why the difference between LPS and PM_nat (or PM_adv) is larger for AT than MART or TRADES (in MART and TRADES, [r1] has found that their loss landscapes are smoother than that regarding AT).
>
> Moreover, the PM_adv and PM_nat assign weights regarding distance from adversarial/natural examples, which rely on slightly different assumptions to measure the importance of data. In principle, when we wish the model to be robust to a particular attack, we suggest using PM_adv in weight assignment. By contrast, if the model should be insensitive to many other attacks, PM_nat might be preferred as its value is calculated using the natural data and the model. The supporting evidence is shown in Table 1: PM_adv is better than PM_nat when defending against PGD attack (the third row) consistently, while this scenario is not true for the AA attack (in the last row, PM_nat might be better than PM_adv).
>
> [r1] Adversarial Weight Perturbation Helps Robust Generalization, NeurIPS, 2020.
> [r2] Maximum Mean Discrepancy Test is Aware of Adversarial Attacks, ICML, 2021.

---

### Official Review · Reviewer_6oSF · 2021-07-16

**Rating:** 7
**Confidence:** 4

**Summary:**

DNNs are susceptible to adversarial attacks, and AT is an efficacious method in combating this challenging issue. In general, AT creates adversarial examples on the fly, and the DNN is trained to overcome their impacts.

The paper mainly focuses on the drawbacks of the reweighted AT training. First, the authors state empirically that the LPSs used in previous works are discrete and path-dependent, unreliable for weight estimation. They then propose a novel metric (PM) that overcomes the drawbacks in LPSs and devise three specifications that capture different geometric properties of the data. Next, a weighting strategy uses PMs to develop a new kind of reweighted AT method called margin-aware instance reweighting learning (MAIL).


**Limitations And Societal Impact:**

The proposed AT method maybe not general to all attacking methods. The main reason is that there is only one attack method used for training. I would be happy to see training the network using other attack methods.

It seems that the GAIRAT, an advanced AT with weighting, does not perform well for the AA attacks in Table 1 and 2. I am curious to know the reason for its inferior results.


**Main Review:**

### Originality:
Although the reweighted AT training has been explored in recent studies, the paper claims the drawbacks of their PGD-based metrics in weight assignment. Accordingly, the authors propose three types of PMs, which can be viewed as the multi-class margin in the probabilistic space of model output. There are some novelties as they address the drawbacks in previous works, and the experimental results look pretty convincing to me.

### Quality:
The paper is motivated by the drawbacks of previous works, and the detailed realizations look technically sound. The experiments are shown on a host of datasets, which are well executed, reproducible, and competitive against state-of-the-arts.

### Clarity:
The paper is well structured and clearly written. They provide the realization details and the adopted hyper-parameters in a clear way.

### Significance:
They concentrate on the weighted AT training, which is a promising methodology that attracts increasing attention recently. The paper discusses the drawbacks of the PGD-based metrics used in previous works, which would make the training procedure unstable and inaccurate. Further, the proposed solution is relatively new and seems like an acceptable improvement over the previous works.


**Time Spent Reviewing:**

3

---

> ### Author Response · Authors · 2021-08-09
> **Response to review questions**
>
> >Q1. The proposed AT method may not be general to all attacking methods. The main reason is that there is only one attack method used for training. I would be happy to see training the network using other attack methods.
>
> A1. Many thanks for your constructive suggestions! In fact, our MAIL can easily be generalized to many other attacks. As an example, we adopted the CW attack with MAIL-AT for training. The experimental results on the CIFAR-10 dataset with ResNet-18 are listed in the following:
>
> |                | NAT   | PGD   | APGD  | CW    | AA    |
> |----------------|-------|-------|-------|-------|-------|
> | MAIL-AT w. PGD | 83.15 | 55.25 | 53.20 | 47.81 | 44.10 |
> | MAIL-AT w. CW  | 82.74 | 54.33 | 52.47 | 52.65 | 41.80 |
>
> As we can see, using the CW method for training would lead to better robustness regarding the CW attacks, but the robustness is relatively low for the PGD-based attacks and the AA attack. It coincides with the previous conclusion that adversarial training is not adaptive [r1].
>
> >Q2. It seems that the GAIRAT, an advanced AT with weighting, does not perform well for the AA attacks in Table 1 and 2. I am curious to know the reason for its inferior results.
>
> A2. The GAIRAT performs poorly on the AA attacks due to the drawbacks of the LPS in geometric measurement, i.e., they are path-dependent and discrete. Accordingly, the assigned weights may make mistakes and thus mislead the model. Although one can try to tune the hyperparameters (such as \lambda) for a convincing result for the PGD attack, it would lead to obfuscated gradients and thus perform poorly on other attacks (e.g., the AA attack).
>
> [r1] SoK: Towards the Science of Security and Privacy in Machine Learning.

---

### Official Review · Reviewer_bxMc · 2021-07-18

**Rating:** 8
**Confidence:** 4

**Summary:**

Reweighting adversarial data has been recently shown to be effective, with larger weights assigned for data closer to the boundaries. Previous work used the LPS to measure the closeness, but the authors suggested that the LPS might not be reliable since they are discrete and path-dependent. To this end, this paper proposed three types of probabilistic margin, which could overcome the drawbacks of the LPS. The probabilistic margins were used to measure the importance of adversarial data, which induced convincing results in experiments.

**Limitations And Societal Impact:**

1.	The drawbacks of LPS stem from the use of the PGD method, but it seems that the proposed LM-PGD can alleviate some of these drawbacks. Therefore, I think LM-PGD would help calculate the LPS, and it would be great if the authors could clarify, i.e., why is the PM better than the LM-PGD?
2.	Computing the margin in the probabilistic space of the model outputs may look convincing. However, I am curious about the results without normalization, i.e., if the proposal only adopted the embedding features (without softmax as normalization) for weight assignment, how much will the model's performance degrade?
3.	Another little question, I think the authors could describe why the constraint of the sum is required in Eq. 3.

**Main Review:**

Instance-reweighted adversarial training assigns unequal importance for adversarial examples, where data close to decision boundaries are vulnerable to be attacked and should be given with large weights. However, the authors stated that the existing methods measuring the closeness (i.e., LPS) are unreliable and experimentally verified their consequence is non-negligible. In my opinion, it might partially explain why GAIRAT cannot work well regarding advanced attack methods such as CW.

This paper proposed the probabilistic margin, which is the multi-class margin measured by the model outputs. The margin is a natural measurement of the closeness in SVM, and it is computed in an embedding space with normalization for the sake of the same scale. The modification is well-motivated and partly overcomes the drawbacks in GAIRAT.

Based on the probabilistic margin, this paper proposed a general framework for instance reweighing learning. MAIL could boost the performance of previous methods (e.g., MART) and can adopt different kinds of probabilistic margins in weighting.

The authors conducted extensive experiments on various datasets and with different probabilistic margins. MAIL not only outperformed the GAIRAT by a large margin for many attacks (e.g., AA) but also achieved comparable results against many state-of-the-art methods (e.g., MART).

**Time Spent Reviewing:**

36

---

> ### Author Response · Authors · 2021-08-09
> **Response to review questions**
>
> >Q1. The drawbacks of LPS stem from using the PGD method, but it seems that the proposed LM-PGD can alleviate some of these drawbacks. Therefore, I think LM-PGD would help calculate the LPS, and it would be great if the authors could clarify, i.e., why is the PM better than the LM-PGD?
>
> A1. Many thanks for your constructive comments! We would like to emphasize that the LM-PGD cannot wholly remove the drawbacks of the LPS; here are the reasons: (1) the path-dependency property exists as the LM-PGD still uses the first-order gradient for optimization; (2) the momentum term and the line-searched learning rate would make the corresponding LPS difficult to be computed since each step may have different length. As a result, we believe that the LM-PGD cannot be directly used in assigning weights, and thus we proposed the probabilistic margin.
>
>
> >Q2. Computing the margin in the probabilistic space of the model outputs may look convincing. However, I am curious about the results without normalization, i.e., if the proposal only adopted the embedding features (without softmax as normalization) for weight assignment, how much will the model's performance degrade?
>
>
> A2. According to your kind suggestion, we have conducted experiments on CIFAR-10 dataset with the backbone being ResNet-18. The experimental results are listed in the following:
>
> |                       | NAT   | PGD   | APGD  | CW    | AA    |
> |-----------------------|-------|-------|-------|-------|-------|
> | MAIL-AT               | 83.15 | 55.25 | 53.20 | 47.81 | 44.10 |
> | MAIL-AT w.o. norm     | 82.87 | 55.14 | 51.20 | 45.70 | 42.58 |
> | MAIL-TRADES           | 81.84 | 53.68 | 52.92 | 52.89 | 50.03 |
> | MAIL-TRADES w.o. norm | 81.45 | 53.40 | 52.19 | 51.22 | 48.63 |
>
> In general, the normalized embedding features would genuinely lead to better results, which experimentally verifies the rationality in using the probabilistic margin (instead of the commonly-used margin) as the geometric measurement.
>
> >Q3. Another little question, I think the authors could describe why the constraint of the sum is required in Eq. 3.
>
> A3. This constraint of the sum is also adopted in previous works [r1]. In general, we wish the risk after weighting is consistent with the original one without weights, and this simple trick of the constraint would help us achieve this goal.
>
> [r1] Geometry-aware Instance-reweighted Adversarial Training (ICLR21).

---

### Decision · Program_Chairs · 2021-09-27

**Decision:**

Accept (Poster)

**Comment:**

The paper studied the reweighting strategy in adversarial training and got 8776 scores. Before rebuttal, only one reviewer had some concerns on the experiments, while the authors provided a successful rebuttal with additional experiments and addressed the reviewer's concerns. Thus, I recommend accepting the paper.